# Impact of Visceral Leishmaniasis on Local Organ Metabolism in Hamsters

**DOI:** 10.3390/metabo12090802

**Published:** 2022-08-27

**Authors:** Mahbobeh Lesani, Camil Gosmanov, Andrea Paun, Michael D. Lewis, Laura-Isobel McCall

**Affiliations:** 1Department of Microbiology and Plant Biology, University of Oklahoma, Norman, OK 73019, USA; 2Department of Chemistry and Biochemistry, University of Oklahoma, Norman, OK 73019, USA; 3Laboratory of Parasitic Diseases, National Institute of Allergy and Infectious Diseases, NIH, Bethesda, MD 20892, USA; 4Department of Infection Biology, London School of Hygiene and Tropical Medicine, London WC1E 7HT, UK; 5Laboratories of Molecular Anthropology and Microbiome Research, University of Oklahoma, Norman, OK 73019, USA

**Keywords:** visceral leishmaniasis, *Leishmania donovani*, metabolites, liver, spleen, gut, GNPS, LC-MS/MS

## Abstract

*Leishmania* is an intracellular parasite with different species pathogenic to humans and causing the disease leishmaniasis. *Leishmania donovani* causes visceral leishmaniasis (VL) that manifests as hepatosplenomegaly, fever, pancytopenia and hypergammaglobulinemia. If left without treatment, VL can cause death, especially in immunocompromised people. Current treatments have often significant adverse effects, and resistance has been reported in some countries. Determining the metabolites perturbed during VL can lead us to find new treatments targeting disease pathogenesis. We therefore compared metabolic perturbation between *L. donovani*-infected and uninfected hamsters across organs (spleen, liver, and gut). Metabolites were extracted, analyzed by liquid chromatography-mass spectrometry, and processed with MZmine and molecular networking to annotate metabolites. We found few metabolites commonly impacted by infection across all three sites, including glycerophospholipids, ceramides, acylcarnitines, peptides, purines and amino acids. In accordance with VL symptoms and parasite tropism, we found a greater overlap of perturbed metabolites between spleen and liver compared to spleen and gut, or liver and gut. Targeting pathways related to these metabolite families would be the next focus that can lead us to find more effective treatments for VL.

## 1. Introduction

Leishmaniasis is one of the neglected tropical diseases, caused by a protozoan parasite, *Leishmania*. There are different human-pathogenic *Leishmania* species, which associate with different disease symptoms. Clinical forms of leishmaniasis are distinguished by infected tissue location and severity, and include cutaneous (CL), mucocutaneous (MCL) and visceral leishmaniasis (VL). VL can be asymptomatic or with symptoms like persistent fever, splenomegaly, hepatomegaly, weight loss, and anemia. Mortality, if untreated, ranges from 10–20% of cases, making recent estimates of yearly 50,000 to 90,000 new VL cases concerning. Most VL cases are found in Southeast Asia, Sudan, Ethiopia and Brazil. *Leishmania donovani* is a species that causes VL in some Asian countries like India, Bangladesh, and Nepal and some African countries like Sudan and Ethiopia [1,2,3,4].

Standard treatments for VL are very limited and include pentavalent antimonials, amphotericin B, pentamidine, miltefosine, and paromomycin, or combination treatments. Apart from miltefosine, all are parenteral. A further concern is the rise of resistance, necessitating increases in dose and treatment duration, leading to worsened adverse effects such as renal damage, ototoxicity, cardiac and liver issues [2,5]. The efficacy of treatment and adverse effects are even more concerning for immunocompromised patients that are infected with *Leishmania* [6]. Finding new treatments with less toxicity and higher efficacy will have a significant impact on the quality of these patients’ life. Modulating host metabolism has recently emerged as a new approach to treat neglected tropical diseases. For example, a previous study on the related parasite *Trypanosoma cruzi* used metabolomics to identify carnitine as a new candidate treatment for acute-stage Chagas disease with efficacy in experimental models of disease [7]. Such an approach could be valuable in the context of leishmaniasis as well.

There have been many studies on metabolite profiling of different species of *Leishmania.* However, most of them have been done in vitro rather than in vivo (e.g., [8,9,10]). A recent study analyzed the metabolome of the liver, spleen, serum, brain, and urine of *L. donovani*-infected mice, and observed alterations in fatty acids, dicarboxylic acids, amino acids, and pyridine [11]. However, this study used gas chromatography–mass spectrometry (GC-MS), an efficient technique to separate non-polar analytes that have masses less than 600 Dalton but less suitable for larger and polar analytes [12,13,14]. Furthermore, previous studies showed that Golden Syrian hamsters (*Mesocricetus auratus*) are better animal models of VL compared to other models for immunopathogenesis and drug development [15]. In hamsters, however, metabolomic studies have been limited to serum lipids. Specifically, Qin et al. (2022) used liquid chromatography—mass spectrometry (LC-MS) to show that glycerophospholipids, α-linoleic acid, and arachidonic acid are the metabolites most affected by infection in the serum [16]. Tissue metabolomic analyses in hamsters using LC-MS are lacking. In this study, we address this gap by analyzing the metabolome of the gut, liver, and spleen from *L. donovani*-infected hamsters by LC-MS. Our findings can lead us to find more effective treatments to increase the tolerance or resistance of the host in VL.

## 2. Materials and Methods

### 2.1. Animals and Infections

Parasite culture, hamster infection, anti-*Leishmania* IgG ELISA and parasite burden quantification were previously performed and reported in [17]. Briefly, 4–6 weeks old Golden Syrian hamsters were infected with metacyclic *L. donovani* strain 1S (MHOM/SD/00/1S-2D) promastigotes (3 × 10^7^) by intracardiac injection. 12–14 weeks post infection, hamsters were euthanized, exsanguinated and perfused with PBS. Tissues were collected from 14–16 hamsters (gut (14 hamsters), spleen (16 hamsters, two samples per animal), and liver (14 hamsters)), snap-frozen and stored at −80 °C until metabolite extraction. The experiment was performed according to the NIH Guide for the Care and Use of Laboratory Animals and under approved protocol LPD 68E.

### 2.2. Sample Preparation for LC-MS/MS

Metabolite extraction started with adding chilled LC-MS-grade water, normalized by sample weight (500 µL water/50 mg sample), and homogenizing in a Tissuelyzer (Qiagen, Hilden, Germany, Tissuelyzer II) for 3 min at 25 Hz with a 5-mm steel ball. Then, to achieve a final concentration of 50% methanol to water, methanol spiked with sulfachloropyridazine (4 µM) was added at 500 µL methanol/50 mg sample. Samples were re-homogenized in a Tissuelyzer for 3 min at 25 Hz. Homogenized tissues were centrifuged with 16,000× *g* for 10 min at 4 °C. The supernatant was collected and dried down in a Speedvac. To extract organic metabolites, dichloromethane: methanol 3:1 spiked with sulfachloropyridazine (2 µM) was added to the pellet and homogenized for 3 min at 25 Hz. Samples were centrifuged at 16,000× *g* for 10 min at 4 °C. The supernatant was air-dried overnight. Both extracts were stored at −80 until LC-MS analysis.

### 2.3. Liquid Chromatography–Tandem Mass Spectrometry

Both extracts were combined by adding 200 µL of 50% methanol spiked with 2 µM sulfadimethoxine as internal standard. Samples were analyzed by LC-MS (Thermo Scientific Vanquish UHPLC system and Q Exactive Plus MS) using a C8 column (0.7 μm, 50 mm × 2.1 mm, 100 Å Kinetex). Instrumental parameters are in Appendix A. MS/MS data were collected in both positive and negative mode.

### 2.4. LC-MS/MS Data Analysis

After converting the raw data to mzXML format using MSConvert software (version 3.0.19014-f9d5b8a3b, Proteowizard, Palo Alto, Santa Clara, CA, USA) [18], mzXML files were processed with MZmine version 2.35 to generate the feature table (Appendix A) [19]. Features that had less than threefold difference to blank were removed. After blank removal, total ion current (TIC) normalization was done in R version 3.6.1 (http://jupyter.org, accessed on 5 December 2019). PCoA plots were generated with TIC normalized data using the Bray–Curtis dissimilarity metric in QIIME2 [20] and visualized with EMPeror [21]. Features were annotated through molecular networking using Global Natural Products Social Molecular Networking (GNPS) (accessed on 23 July 2021) by applying an in-house automated script (https://github.com/camilgosmanov/GNPS, (accessed on 19 November 2021) to collect annotations and mirror plots from GNPS (Appendix A) [22,23] and were visualized with Cytoscape [24]. Venn diagrams to show common or specific metabolites were generated in R. Random forest classification analysis [25] was performed with 1000 trees (using the random forest R package) to determine features that contribute the most differences between uninfected and infected tissues in Jupyter notebook (http://jupyter.org, accessed on 5 December 2019), followed by Wilcoxon test. Adonis tests were performed in QIIME 2 [20]. Perturbed chemical families in the three organs were found by applying PALS (Pathway Activity Level Scoring) for family members ≥10, *p* < 0.05 [26]. Glycerophosphocholines were identified by searching the data for the PC diagnostic peaks of *m/z* 184.0739, 125.0004, 104.1075, and 86.09697 [27] using MassQL, https://github.com/mwang87/MassQueryLanguage (accessed on 27 February 2022) [28]. The hierarchical clustering heatmap were generated through Versatile matrix visualization and analysis (MORPHEUS) (https://software.broadinstitute.org/morpheus/) (accessed on 13 March 2022).

## 3. Results

VL is associated with localized parasite colonization in the liver and spleen. Recently, the gut has also emerged as a target organ [17]. The immune response to the parasite differs between sites [29]. However, it is incompletely understood how infection reshapes the metabolome differentially across infection sites. To address this gap, we acquired LC-MS/MS data from the liver, gut (duodenum) and spleen of *L. donovani*-infected hamsters at 12 to 14 weeks post-infection and from uninfected controls. The overall metabolome was most perturbed in the liver (PERMANOVA R^2^ (positive mode) = 0.309, R^2^ (negative mode) = 0.326, *p* < 0.01). In contrast, the effect of infection was more minor in the spleen (PERMANOVA R^2^ (positive mode) = 0.137, R^2^ (negative mode) = 0.270, *p* < 0.01). With regard to the gut, we only observed significant changes in negative mode (PERMANOVA R^2^ (positive mode) = 0.134, *p* > 0.05; R^2^ (negative mode) = 0.137, *p* < 0.05) (Figure 1A–C and Appendix A). No significant relationship between antibody responses (anti-*Leishmania* IgG ELISA) and overall infected animal metabolome were observed in gut, liver, and spleen (positive mode, PERMANOVA *p* > 0.05). Overall metabolome and parasite load in the liver likewise did not show a significant relationship (positive mode, PERMANOVA *p* > 0.05), in contrast with the spleen (PERMANOVA (positive mode) *p* < 0.05 R^2^ = 0.159). No parasite burden data was available for gut samples.

Next, we used random forest machine learning analysis to first determine the proportion of metabolites commonly perturbed by infection across tissues, and second, the specific metabolites and metabolic pathways that are most perturbed by infection. Using this approach, we identified more metabolites specifically perturbed in the liver and spleen than the gut for both data acquisition modes. Furthermore, the gut had the lowest percentage of perturbed metabolites compared to the other two organs, in both modes (*p* < 0.05, Fisher’s exact test). In contrast, this percentage was comparable between liver and spleen (*p* > 0.05, Fisher’s exact test). There were more metabolites commonly perturbed between liver and spleen compared to liver-gut or spleen-gut comparisons (Fisher’s exact test, *p* < 0.01) for both modes. Of the infection-perturbed metabolites overlapping between two organs, across both data acquisition modes, 51% had the same direction of change. Few perturbed metabolites were in common across all three organs (17 features across both modes), with 35.3% (6/17) of them showing the same direction of change in response to infection in all three organs (Figure 1D and Appendix A).

We performed feature-based molecular networking (FBMN) to annotate these differential metabolites (Appendix A) [22]. Kynurenine in particular was increased across all three organs, with a higher fold change to uninfected samples in liver and spleen compared to gut (Figure 1E). Kynurenine normalized peak area in infected samples was not correlated with parasite burden in liver and spleen (Appendix A), nor with antibody responses (anti-*Leishmania* IgG ELISA) in spleen, liver and gut (Appendix A). Riboflavin is an example of location-specific perturbed metabolites, being increased by infection in the liver only (Figure 1F). Similar to kynurenine, riboflavin normalized peak area in infected samples was not correlated with parasite load in liver and spleen (Appendix A) nor with antibody responses in liver, spleen and gut (Appendix A). Kynurenine helps to regulate the balance between activation and inhibition of the immune system [30]. Riboflavin is a vitamin and a precursor in coenzyme biosynthesis, with antioxidant, immunomodulatory and antimicrobial activity [31].

Some metabolite families had high representation as infection-perturbed metabolites, including glycerophosphocholines, phosphoethanolamines, fatty acyls, acylcarnitines, purines and amino acids. To investigate the impact of infection at the chemical family level, we applied PALS (Pathway Activity Level Scoring) [26] separately on each organ. The greatest number of perturbed chemical families was observed in the liver and the least number of perturbed chemical families in the spleen (Family members ≥ 10, *p* < 0.05, Appendix A).

Families that were perturbed by infection in all three organs include glycerophospholipids, ceramides, acylcarnitines, peptides, purines, amino acids, lipids, and lipid-like molecules (Figure 2). Most chemical families showed a mixed pattern of change, with some family members increased and some decreased by infection (Figure 2A–S). In contrast, most ceramides and acylcarnitines were increased in the spleen (Figure 2D,G). Acylcarnitines were also almost all increased in the liver (Figure 2H).

Glycerophosphocholines were the most prevalent infection-perturbed chemical family, so we focused on this family in greater detail. We used the characteristic phosphocholine head group MS2 fragmentation pattern [27] to collect all glycerophosphocholines in our dataset. Hierarchical clustering of glycerophosphocholine peak areas successfully separated infected and uninfected samples in liver and spleen (except for one infected sample, Figure 3A,B), while in the gut such clustering was not observed (Figure 3C).

There were three perturbed chemical families in common between spleen and liver (carbohydrates, glycerophosphoethanolamines, and fatty acids, Figure 4A–F). Carbohydrates and fatty acyls were mostly decreased by infection in the spleen, whereas a mixed pattern was observed in the liver. In the case of glycerophosphoethanolamines, most of the individual metabolite features showed a clear pattern of increasing by infection. On the other hand, liver and gut had four perturbed families in common (Figure 4G–N). Most of these families showed a mixed effect of infection within the family, except for amino acids and peptides, which were mostly increased by infection in liver. Spleen and gut had two families specifically perturbed in these organs. Purines and nucleosides were increased by infection in the spleen, whereas bile acids were mainly decreased in the gut (Figure 5A–D).

## 4. Discussion

Overall, we found that *L. donovani* infection caused metabolic perturbations in all three analyzed organs (liver, spleen, gut) in terms of overall metabolism (Figure 1 and Appendix A), and at the individual metabolite (Appendix A) and metabolic family level (Figure 2, Figure 3, Figure 4 and Figure 5, Appendix A). Importantly, many responses were location-specific. Studying organ-specific metabolic alterations help us to understand how these changes can lead to systemic disorders. For example, in this study riboflavin was increased by infection only in the liver. Riboflavin has an important role in immune function and responses like neutrophil and macrophage activation, also having an anti-inflammatory role [32]. Trypanosomatids like *Leishmania* are not able to biosynthesize riboflavin and take it from their environment. Trypanosomatids needs riboflavin as precursors for flavin mononucleotide (FMN) and flavin adenine dinucleotide (FAD), and other essential cofactors for redox center enzymes [33]. Thus, elevated riboflavin could be benefiting the parasite. On the other hand, using riboflavin and ultraviolet light decreased infectivity of *L. donovani* in plasma and whole blood [34,35]. Beyond effects directly on the parasite, several in vivo studies showed that riboflavin treatment can reduce liver damage by decreasing oxidative stress biomarkers like malondialdehyde and increasing glutathione (GSH) [36,37,38,39]. Thus, the elevated riboflavin levels we observed could be affecting both parasite and host aspects of VL. Previous studies in mice demonstrated that the outcome of *L. donovani* infection is not the same in the liver and the spleen, with different immune responses in these two organs. The liver kills *Leishmania* efficiently by forming hepatic granulomas; in contrast, the spleen is not able to control amastigotes growth and shows inflammation during the chronic stage [29]. On the other hand, hamsters infected with *L. donovani* showed progression of infection in both spleen and liver, but different immune gene expression in these two organs [17,40]. Finding location-specific metabolites that are perturbed by infection may link local metabolism to location-specific immune responses.

In contrast, several commonalities were observed across organs and could be targets for systemic intervention. For example, kynurenine was increased by infection in all three organs, with the highest increase in the liver. Previous studies showed that kynurenine production from tryptophan is increased by bacterial, parasitic, and viral infection [41,42,43]. Infection by the related protozoan parasite *Trypanosoma cruzi*, for example, increased kynurenine in the plasma and the heart [43]. Kynurenine is anti-inflammatory [41,44,45] and regulates naïve T cell differentiation to regulatory T (Treg) cells [46]. Thus, kynurenine may be contributing to parasite persistence while limiting tissue damage.

Likewise, purines, peptides, acylcarnitines, glycerophospholipids, and ceramide families were perturbed by infection in all three organs. The latter two changes might be caused by *Leishmania* salvaging phospholipids and sphingolipids [47], or by detection of parasite-derived lipids. Previous studies on *Trypanosoma cruzi* showed elevation of acylcarnitine C20:4 in the esophagus and small intestine during acute-stage infection [7], and likewise we saw elevated acylcarnitines in this study. This may reflect infection-induced alterations in energy metabolism during kinetoplastid infection [48]. Purines and pyrimidines are salvaged by *Leishmania.* Compounds targeting these pathways have been successful as parasite growth inhibitors in vitro and in mouse models of acute Chagas disease [49,50]. In dogs, the inhibitor of purine metabolism, allopurinol, was sufficient to improve disease symptoms, even in the absence of effects on parasite load [51]. This suggests a causal link between our observations and disease severity. *Leishmania* also scavenge many amino acids such as phenylalanine, leucine, and valine. Most amino acids increase promastigotes growth. *Leishmania* also scavenge vitamins (biopterin, folic acid, riboflavin, and lipoic acid) and cofactors [52]. We observed the greatest decrease of purines and amino acids in the spleen compared to liver and gut. Our findings contrast with a prior study in mice where purine and pyrimidine metabolic pathways in liver and spleen did not show significant differences upon *L. donovani* infection [11]. This discrepancy may be caused by the higher parasite load in the hamster model [17]. However, our results are consistent with these known parasite scavenging activities.

Our analysis also showed that lipids and lipid-like molecules are the most commonly perturbed metabolites and metabolic families in response to infection. These observations concur with a recent study on the serum of hamsters infected with *L. donovani* [16] and on cutaneous lesions caused by *Leishmania major* in mice [53]. Previous studies indicated reduction of cholesterol and increased triglycerides in the serum of infected individuals [54,55,56], which may be caused by retention of cholesterol into the parasitophorous vacuole in macrophages [57]. Mice with high fat and high cholesterol diets showed reduced burden of *L. donovani* amastigotes in liver and spleen [58]. A previous study on dogs infected with *Leishmania infantum*, another agent causing VL, showed elevation of post-prandial bile acids concentration in serum [59]. These results suggest a possible causal role in VL pathogenesis of the lipid alterations we observed. Although we detected many perturbed glycerophosphocholines, we did not detect lysophosphatidylcholine or lysophosphatidic acid that can contribute to fibrosis during visceral leishmaniasis [60,61,62]. This could be due to the rapid turnover of lysophosphatidic acid [63].

Further work will be necessary to determine why these metabolites or chemical families are perturbed by infection. One possibility is that elevated metabolites are parasite-derived. Indeed, *Leishmania* can also produce many phospholipids [47] and the *Leishmania* genome has several putative kynurenine pathway genes [64,65].These metabolites and metabolite families have the potential to be biomarkers for VL diagnosis or can lead us to find more effective treatments to increase the tolerance or resistance of the host in VL.

## Figures and Tables

**Figure 1 metabolites-12-00802-f001:**
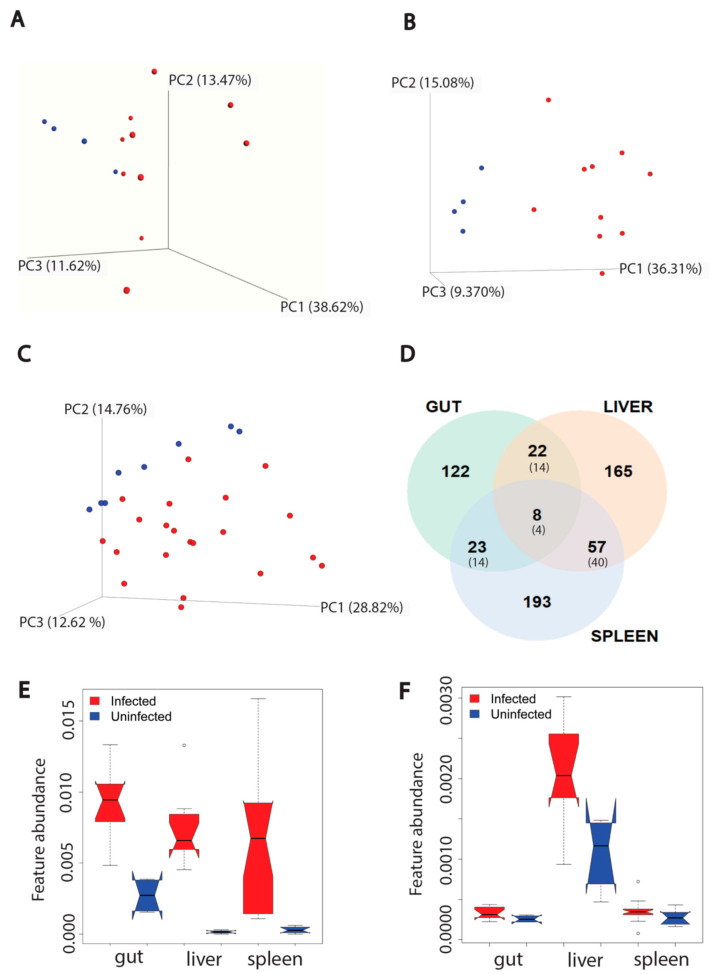
Effect of *L. donovani* infection on overall metabolism of host organs (gut (*n* = 4 uninfected and *n* = 10 infected), liver (*n* = 4 uninfected and *n* = 10 infected), spleen (*n* = 8 uninfected and *n* = 21 infected, two samples per animal)) in positive mode. (**A**) PCoA analysis of extracted metabolites from infected and uninfected gut samples (PERMANOVA R^2^ = 0.134, *p*-value > 0.05). Each sphere represented one sample. Red color represented infected samples and blue color, uninfected samples. (**B**) PCoA analysis of extracted metabolites from infected and uninfected liver samples (PERMANOVA R^2^ = 0.309, *p*-value < 0.01). (**C**) PCoA analysis of extracted metabolites from infected and infected spleen samples (PERMANOVA R^2^ = 0.137, *p*-value < 0.01). (**D**) Venn diagram representing the number of metabolites perturbed by infection at each organ. Smaller font numbers in parentheses show the number of metabolites commonly perturbed between indicated organs and that have the same direction of change in response to infection. (**E**) Kynurenine metabolite increased by infection in three organs. ∘, Wilcoxon rank-sum test with FDR correction, *p*-value < 0.01. (**F**) Riboflavin increased by infection only in liver. ∘, Wilcoxon rank-sum test with FDR correction, *p*-value < 0.01 in liver. Feature abundance is quantified as TIC-normalized peak area.

**Figure 2 metabolites-12-00802-f002:**
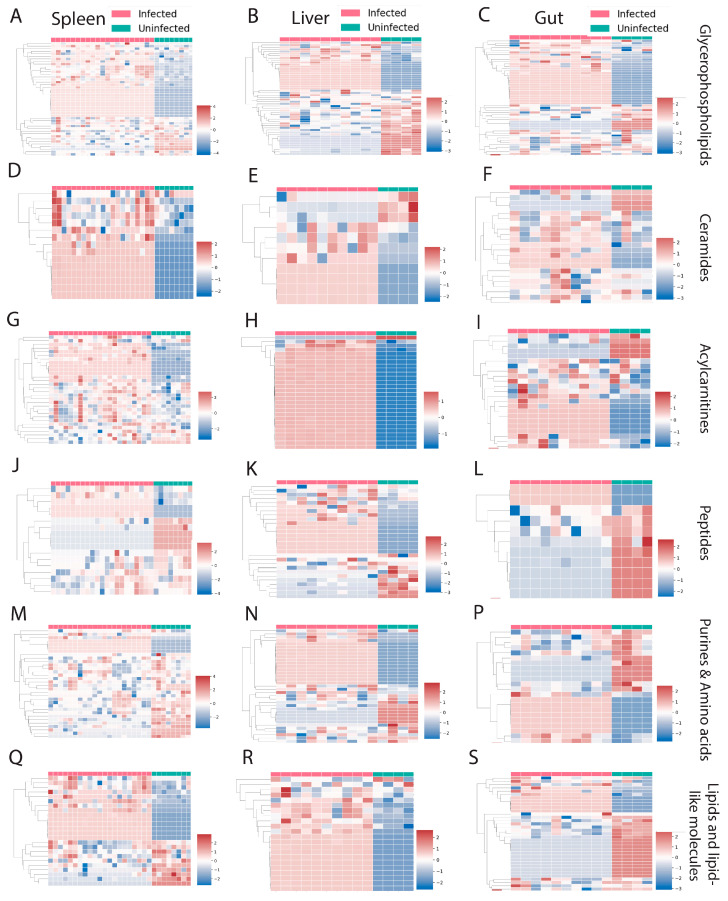
Perturbed chemical families that are in common between three organs in both negative and positive mode. (**A**–**C**) Glycerophospholipids. (**D**–**F**) Ceramides. (**G**–**I**) Acylcarnitines. (**J**–**L**) Peptides. (**M**–**P**) Purines and amino acids. (**Q**–**S**) Lipid and lipid-like molecules. Infected samples are shown in red and uninfected samples in green. Each column represents one sample, and each row represents one metabolite feature. Scale shows zero mean and unit variance (higher than zero mean = red and lower than zero mean = blue color) for each row.

**Figure 3 metabolites-12-00802-f003:**
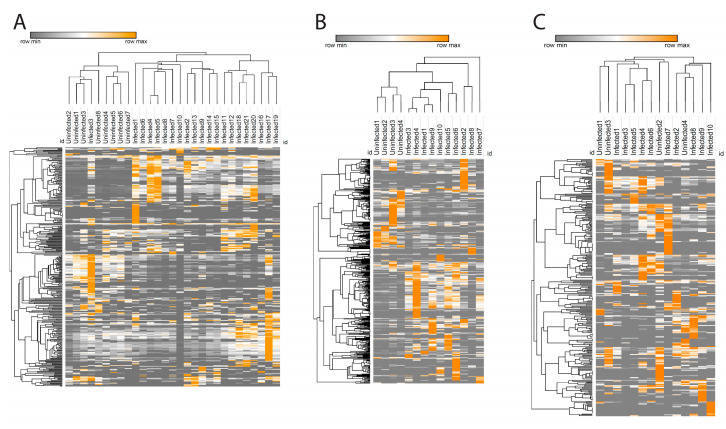
Glycerophosphocholines heat map in spleen (**A**), liver (**B**), and gut (**C**). Each column reperesent one sample and each row represents one metabolite feature from the glycerophosphocholine family. Color scheme shows the minimum (gray) and maximum (orange) values in each row.

**Figure 4 metabolites-12-00802-f004:**
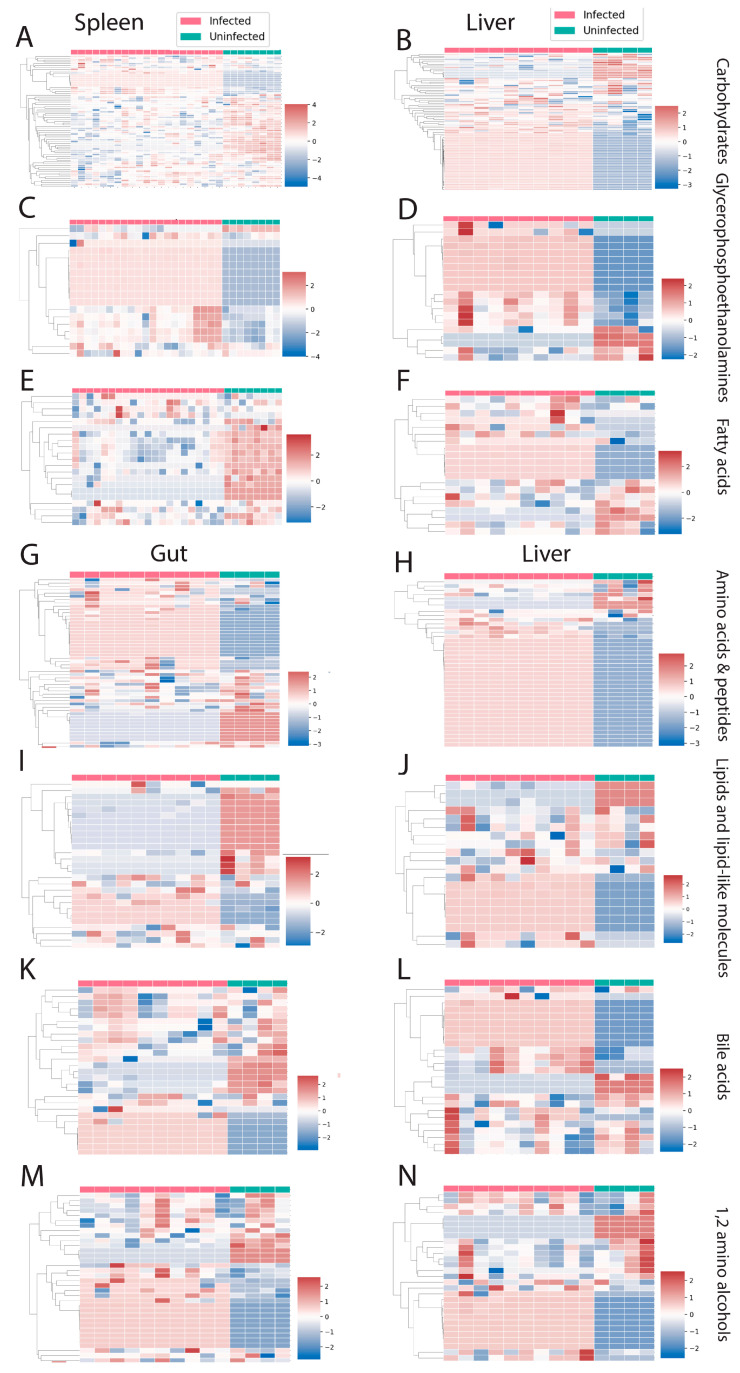
Perturbed chemical families in common between two organs. (**A**–**F**) Perturbed chemical families in common between liver and spleen (carbohydrates, glycerophosphoethanolamines, and fatty acids). (**G**–**N**) Perturbed chemical families in common between gut and liver (amino acids and peptides, lipid and lipid-like molecules, bile acids, 1,2-amino alcohols). Infected samples are shown in red and uninfected samples in green. Each column represents one sample, and each row represents one metabolite feature. Scale shows zero mean and unit variance (higher than zero mean = red and lower than zero mean = blue color) at each row.

**Figure 5 metabolites-12-00802-f005:**
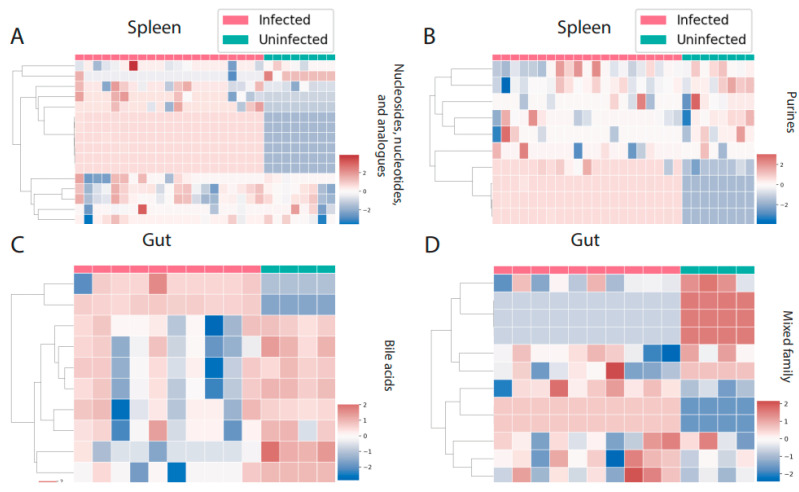
Perturbed chemical families specific to one organ. (**A**,**B**) Nucleosides, nucleotides, and analogues and purines are the perturbed chemical families that are specific to the spleen. (**C**,**D**) Bile acids and a mixed family are the two families that were perturbed by infection only in the gut. Infected samples are shown in red and uninfected samples in green. Each column represents one sample, and each row represents one metabolite feature. Scale shows zero mean and unit variance (higher than zero mean = red and lower than zero mean = blue color) at each row.

## Data Availability

Data were uploaded to MassIVE (massive.ucsd.edu (accessed on 23 July 2021), accession numbers MSV000085991 (positive mode) and MSV000085990 (negative mode)). Molecular networking jobs are accessible through these links: Positive mode: https://gnps.ucsd.edu/ProteoSAFe/status.jsp?task=a41e05f0699e45328675aac49f667753 (accessed on 23 July 2021); Negative mode: https://gnps.ucsd.edu/ProteoSAFe/status.jsp?task=abc2bd81f18c461a9b5e43c8cee83a02 (accessed on 23 July 2021).

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
