# Peer review of "Impact of Visceral Leishmaniasis on Local Organ Metabolism in Hamsters"

_metabolites, 2022, doi:10.3390/metabo12090802_

Round 1
Reviewer 1 Report
The study is worthy of publication in Metabolites and of general interest, however, the authors should address the following points in order to interest this journal’s broad readership:
The manuscript is excessively descriptive and presents few arguments pointing to mechanistic insights afforded by this study’s experimental approach. The findings regarding kynurenine and riboflavin metabolites merit a more articulate and referenced discussion as a service to the scientific community that studies VL and the metabolism of infectious diseases.
Line 206: the affirmation needs a reference(s).
The number of parasites employed to infect hamsters seem exceedingly high. The authors should comment and justify their protocol.
The data in the heat maps are for individual samples, but why do they not cluster with or indicate correlations between metabolic profiles and parasite loads and/or clinical manifestations in hamster hosts? Hamsters are indeed a better model than mice to study VL, but one of the advantages of this model is that the outcomes of infection mimic what is seen in humans and dogs, where they range from asymptomatic infections to overt clinical manifestations of VL and even death. If spleen and liver specimens are available, the study would benefit greatly by adding data for parasite burdens; this is very easy so accomplish with qRT-PCRs of samples and parasite-specific primers published in the literature, without great delays or expenses. This exercise could elucidate mechanisms of evolution to disease or asymptomatic infections, as well as indicate a useful biomarker of disease severity.
Bile acids, the metabolism of which in this study was perturbed in guts by infection with L. donovani, are known to activate the inflammasome and inhibit antibody responses of humans to vaccines (Hagan et al., Antibiotics-Driven Gut Microbiome Perturbation Alters Immunity to Vaccines in Humans, Cell, 2019, doi: 10.1016/j.cell.2019.08.010). Does the data indicate a correlation between these metabolic profiles and levels of L. donovani-specific antibodies? If sera are available from these animals, the study would benefit greatly by adding this data: this is also very easy to accomplish without great delays or expenses. This exercise, together with data from the previously suggested exercise, could elucidate mechanisms of susceptibility to develop clinical manifestations in infections with L. donovani and explain the different outcomes of humoral immune responses in this infection.
Glycerophosphocholines are precursors of lysophosphatidic acid (LPA), an important signaling molecule that is involved in organ fibrosis, a very common histopathological finding in hosts presenting with clinically manifest VL (e.g., re: Silva LC, et al. Canine visceral leishmaniasis as a systemic fibrotic disease Int J Exp Pathol. 2013. doi: 10.1111/iep.12010). Chemist colleagues inform this reviewer that turnover of LPA is very high and that this metabolite must be analyzed by target methods, probably explaining why was it not detected. Given the prominence of this group of compounds in the metabolome of infected hamsters, this aspect is worth discussing.
Reviewer 2 Report
The authors present a study with the aim to determine the metabolites perturbed during Visceral Leishmaniosis (VL). To this end, they compared metabolic perturbation between hamster experimentally infected with L. donovani and uninfected hamsters in the spleen, liver, and gut.
This study is pertinent because it provides information that may be useful to advance in the investigation of new drugs targeting the pathogenesis disease for the leishmaniasis treatment.
Leishmaniasis is a neglected tropical disease, it is endemic in more than 98 countries around the world and the VL affects 50,000 and 90,000 people a year. Nowadays, the treatments for VL are still very limited and it is essential to find new therapeutic strategies more accessible, easier to administrate, with less toxicity and higher efficacy. Therefore, studies in this field are needed.
The manuscript is generally well written and presented. I suggest the following modifications and considerations:
Line 60. Please write the meaning of the acronym GC-MS the first time that it is written in the text.
Line 65. Where it is written “Quin et al.,” Write the year in brackets.
Line 65. Please write the meaning of the acronym LC-MS the first time that it is written in the text.
Lines 69-72. Where it is written “We found that…” These sentences are not relevant to include in the background, it is part of the results.
Line 89. Why do you analyse the double of spleen samples? Could you explain it in the manuscript.
Line 249. How many hamsters did you infect for this study?
Line 258. Please, add the information about the tissuelyzer model.
Line 278. Please, include the version of R software that you used.
Line 281. After “Global Natural Products Social Molecular Networking”, write in brackets GNPS.
References. All the scientific names in the references should be in italics. E.g., line 353 write “Leishmania” as Leishmania. Line 356 “Leishmania donovani” as Leishmania donovani etc.
Round 2
Reviewer 1 Report
The authors present new data and references to enrich their discussion. The manuscript is improved and can be published as is. it will interest the scientific community in general as well as immunoparasitologists.